# The Effects of Chronic Consumption of Lipid-Rich and Delipidated Bovine Dairy Milk on Brown Adipose Tissue Volume in Wild-Type Mice

**DOI:** 10.3390/nu13124266

**Published:** 2021-11-26

**Authors:** Zachary J. D’Alonzo, John C. L. Mamo, Liam T. Graneri, Ryusuke Takechi, Virginie Lam

**Affiliations:** 1Curtin Health Innovation Research Institute, Faculty of Health Sciences, Curtin University, Perth, WA 6845, Australia; Zachary.dalonzo@postgrad.curtin.edu.au (Z.J.D.); J.Mamo@curtin.edu.au (J.C.L.M.); Liam.graneri@postgrad.curtin.edu.au (L.T.G.); R.Takechi@curtin.edu.au (R.T.); 2Curtin Medical School, Faculty of Health Sciences, Curtin University, Perth, WA 6845, Australia; 3School of Population Health, Faculty of Health Sciences, Curtin University, Perth, WA 6845, Australia

**Keywords:** milk, hypercaloric diet, brown adipose tissue, body composition

## Abstract

Brown adipose tissue (BAT) activation is associated with increased energy expenditure by inducing non-shivering thermogenesis. The ingestion of a milk fat globule membrane (MFGM) supplement and a high calorie diet are reported gateways into BAT activation. However, little is known about the effect of the MFGM and high calorie diets on BAT volume. To gain insight into this, mice were maintained on a high-fat (HF) or low-fat (LF) diet in conjunction with either full-cream (FC) or skim bovine dairy milk (BDM). After being maintained on their respective diets for 13 weeks, their body composition, including BAT volume, was measured using X-ray microtomography. A high calorie diet resulted in an increase in the BAT volume and mice consuming an HF diet in conjunction with FC BDM had a significantly greater BAT volume than all the other groups. Conversely, mice consuming an HF diet in addition to skim milk had a lower BAT volume compared to the HF control. The data presented suggest that the consumption of a high calorie diet in conjunction with FC BDM increases the BAT volume in wild-type mice. This study may provide valuable insight into future studies investigating BAT volume and BAT activity in relation to environmental factors, including diet.

## 1. Introduction

Mammals regulate heat production through two mechanisms known as shivering and non-shivering thermogenesis (NST) [1]. Shivering thermogenesis defines the conversion of chemical energy into heat through the generation of a contraction in specific muscle groups in response to a cold stimulus. On the contrary, NST defines the biochemical process of heat generation that does not induce shivering and is considered a more efficient method of thermoregulation. Brown adipose tissue (BAT) is considered to be the primary site of NST, where NST takes place by inducing rapid intracellular lipolysis by the activation of adipose triglyceride lipase and hormone sensitive lipase [2,3,4]. This ultimately results in the release of fatty acids by lipid droplets that, thereafter, can be converted into heat via uncoupling protein 1 (UCP1) in mitochondria [5,6]. Conversely, white adipose tissue (WAT) has the opposite role and stores excessive energy as triglycerides [7].

Historically, BAT had been thought to have little significance in adult humans [7]. However, within the past two decades, a number of studies have reported that a higher BAT mass is associated with a lower BMI [8,9]. Furthermore, the activation of BAT is increasingly recognised as an important process for BAT’s energy utilisation. Whilst non-activated BAT has metabolic activity comparable to that of WAT [10], activated BAT has increased thermogenic capabilities, consequentially resulting in an increased energy expenditure and may, therefore, contribute to fat loss [11]. Additionally, evidence suggests that activated BAT has beneficial metabolic effects and reportedly regulates blood glucose levels in type 2 diabetes patients [12].

BAT can be activated by cold exposure or induced by the ingestion of a meal (diet-induced thermogenesis (DIT)). Recent evidence suggests that the milk fat globule membrane (MFGM), the membrane that encompasses milk fat and allows for the efficient distribution of milk lipids, can attenuate body weight gain via the activation of BAT in mice maintained on a high-fat diet [13]. Mice that were fed high-fat chow in addition to a MFGM-rich infant supplement had reduced weight gain of up to 34.82%, which was positively associated with the quantity of MFGM consumption compared to the mice fed only with high-fat chow [13]. The treatment of mice with the MFGM led to an upregulation of UCP1 expression in BAT, indicative of a significantly enhanced conversion of triglycerides into heat. Moreover, the MFGM and phosphatidylcholine were shown to accelerate the conversion of white to beige adipocytes that have thermoregulatory abilities comparable to BAT [14].

Bovine dairy milk (BDM) is a major source of the MFGM and is consumed by many people every day, globally. It is a cheap commodity and is readily available to most people worldwide. However, no studies to date have investigated the effects of commercially available BDM on BAT volume and weight. As BDM is such an abundant product, the potential for BDM to have an effect on BAT volume could have significant implications on the suppression of obesity and associated diseases. Thus, in the present study we evaluated the effect of full cream BDM on BAT and WAT abundance in comparison with delipidated, skim BDM in C57BL/6J mice, and how BDM fat can affect the production of BAT and WAT in mice maintained concurrently on a high- or low-fat diet.

## 2. Materials and Methods

### 2.1. Animals and Dietary Intervention

Young adult wild-type C57BL/6J male mice were purchased from Animal Resources Centre (Canning Vale, WA, Australia) at 5 weeks of age. Following a 1-week acclimatisation period, mice were randomly separated into one of 5 dietary intervention groups (n = 5 per group). The low-fat control group was given water with a standard maintenance chow (LF) (AIN-93M, Specialty Feeds, Glen Forrest, WA, Australia). The high-fat control group was given water with a high-fat chow (HF) (SF07, Specialty Feeds, Glen Forrest, WA, Australia). Two additional groups were maintained on a 20% full cream milk solution diluted in water with one group given the low-fat diet (LF+FC) and one group receiving the high-fat chow (HF+FC). The final group was given 20% skim milk solution diluted in water in addition to the high-fat chow (HF+Skim). See Appendix A for the macronutrient composition for each intervention. Each group was sacrificed 13 weeks after commencing the dietary intervention. Mice were housed in groups of 5 in ventilated cages with a 12-h light/dark cycle under controlled air pressure and temperature (21 °C). All groups had ad libitum access to food and drink. This study was approved by the Curtin Animal Ethics Committee (AEC Approval No. 2018-03).

### 2.2. In Vivo Body Fat Composition Analysis

After 13 weeks of dietary intervention, body composition was measured using Skyscan high resolution in vivo X-ray microtomography (Bruker, Billerica, MA, USA) at the Centre for Microscopy, Characterisation and Analysis (Harry Perkins Institute, Nedlands, WA, Australia) as previously described [15]. Briefly, mice were anesthetised with isofluorane gas, and the thoracic and abdominal areas of each mouse was imaged with 40 kV and 383 µA intensity. To limit noise and artefacts, smoothing was set to 2 and ring artefact reduction was set to 6. To optimise contrast, beam hardening correction was set to 20%. Total body fat percentage, subscapular BAT and total body WAT volumes were calculated through tomography intensity grading of 3D X-ray shadow projection reconstructions using 3D rendering software CT-Analyser (Bruker).

### 2.3. Statistical Analysis

All data are displayed as mean ± SEM. All data were analysed with one-way analysis of variance (ANOVA) followed by Fisher’s least significant difference (LSD) post hoc test for multiple comparisons (GraphPad Prism 9, San Diego, CA, USA). Statistical significance was detected at *p* < 0.05.

## 3. Results

### 3.1. Food and Liquid Consumption

The average daily liquid and food intake per mouse was measured and is presented in Figure 1. The food and liquid intake were used to determine the total cumulative energy from macronutrients, lipids, carbohydrates and proteins over the 13-week period. The LF and HF control groups consumed similar liquid quantities and significantly less compared to all the milk groups (Figure 1A). The HF+FC and HF+Skim groups consumed similar liquid quantities, which were slightly higher than the control groups. The LF+FC group consumed the greatest quantity of liquid, and this was significantly higher than all the other groups. Food consumption was similar across all the groups with only the HF groups being slightly, but significantly, higher than the LF+FC group (Figure 1B).

Both the HF+Skim and HF+FC groups had a significantly higher total energy intake than the HF group and the LF groups, with the HF+Skim group consuming the most energy, followed by the HF+FC group (Figure 1C). All the HF groups had a similar and significantly higher cumulative energy intake from lipids as a result of the ingestion of a high-fat diet (Figure 1D). The LF and LF+FC groups had the highest cumulative energy intake from carbohydrates, which were significantly higher than all the HF groups except for the HF+Skim group (Figure 1E). All the groups had a similar energy intake from protein as all the food and milk solutions contained adequate amounts of protein (Figure 1F). However, the HF+Skim group maintained a slightly higher level of protein intake than the other groups, which was significantly higher than the LF group.

### 3.2. Dietary Intervention Effect on Body Composition

The mass and body composition of the mice are depicted in Figure 2. All the HF groups had a significantly higher body mass compared to all the other LF groups (Figure 2A). The body WAT volume in the LF group was the lowest, followed by the LF+FC group, both of which were significantly lower than all the other groups receiving HF (Figure 2B). Despite the HF+Skim and HF+FC groups having a similar body mass, the HF+FC group had a significantly higher body WAT volume than the HF+Skim group. The HF group had the lowest body mass out of the HF groups, yet significantly higher than the LF or LF+FC groups. The consumption of HF alone or FC milk alone did not significantly increase the BAT volume compared to the mice maintained on LF (Figure 2C). However, the combination of HF+FC significantly increased the BAT volume compared to all the other groups (see Appendix A). The synergistic effects of HF and FC milk intake was not realised when the mice received HF and Skim milk.

The ratio of BAT volume per gram of body weight was used to clearly describe the BAT measured relative to the size of an individual mouse (Figure 2D). Relative to the total body mass, the mice maintained on HF+FC showed a significantly higher BAT volume compared to all the other groups.

## 4. Discussion

BAT is an increasingly researched organ that has known roles in NST and, until recently, was thought to be physiologically irrelevant in the adult human. However, recent research has shown that environmental factors such as cold temperature and certain diets can influence adult BAT activity. Therefore, in this study, we report the effects of the chronic intake of BDM and a hypercaloric HF diet on BAT volume in young adult wild-type mice.

The intervention with a diet enriched in fat (HF) for 13 weeks almost doubled the BAT volume in mice compared to the mice receiving low-fat control chow (LF). Similar observations have been reported by previous clinical and pre-clinical studies, which identified a significant increase in BAT activity after chronic hypercaloric feeding and even after a single hypercaloric meal [16,17,18]. However, these studies did not determine the volume of BAT. Thus, our study, for the first time, demonstrates that high calorie HF diets increase BAT volume.

The mice receiving an HF diet in conjunction with full cream bovine dairy milk (HF+FC) displayed a significantly higher net BAT volume as well as the BAT proportion per total body mass, compared to every other group in the study. Despite consuming an almost identical number of calories and macronutrients over the study period, the mice in the HF diet in conjunction with delipidated skim milk (HF+Skim) group showed a significantly lower BAT volume compared to the HF+FC mice. Similarly, the mice that consumed only FC BDM without a hypercaloric HF diet (LF+FC group) showed a significantly lower BAT volume compared to the mice in the HF+FC group. The combination of FC BDM and the HF diet not only introduced additional lipids to a lipid-rich diet but was also associated with an influx of proteins and phospholipids that form the MFGM. Although we did not analyse BAT activation following the respective interventions, it is known that MFGM supplementation in mice increases the activity of BAT [13,14]. This study suggests that the ingestion of FC BDM in conjunction with a high-fat diet may result in an increased MFGM, which may consequently increase the BAT volume.

Interestingly, both FC and skim milk in combination with an HF diet showed a significant increase in WAT volume, whilst only the mice with HF+FC showed a BAT volume increase. This suggests that the intake of FC milk, but not skim milk, with an HF diet may promote the browning of WAT. Irrespective of the WAT volume, studies suggest that certain nutritional interventions and exercise regimes may promote the browning of WAT to produce beige adipocytes to modulate energy expenditure [19,20,21]. These studies also suggest that the browning of WAT typically coincides with increases in BAT activity. Of particular importance, Li et al. established that the MFGM may induce the browning of WAT alongside BAT activation [13,14]. Therefore, the increase in BAT volume in the HF+FC mice suggests that browning may be taking place in the WAT of this intervention group.

## 5. Conclusions

In this study, we have shown that a high-fat, hypercaloric diet, together with FC BDM, increases the BAT volume in wild-type mice. Further intervention studies with BDM will be required in order to identify the full impact on BAT activity, WAT browning and energy expenditure in human subjects.

## Figures and Tables

**Figure 1 nutrients-13-04266-f001:**
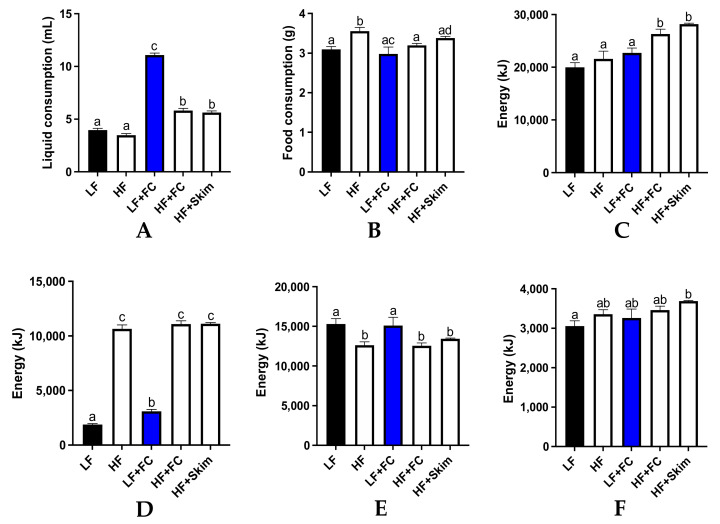
Liquid consumption (**A**) and food consumption (**B**) was measured daily per mouse over the 13-week interventions. Total energy consumption (**C**), total energy from lipids (**D**), total energy from carbohydrates (**E**) and total energy from protein (**F**) was also measured following 13-week interventions. Energy consumptions are cumulative following the 13-week intervention. Groups with unlike letters were significantly different (*p* < 0.05).

**Figure 2 nutrients-13-04266-f002:**
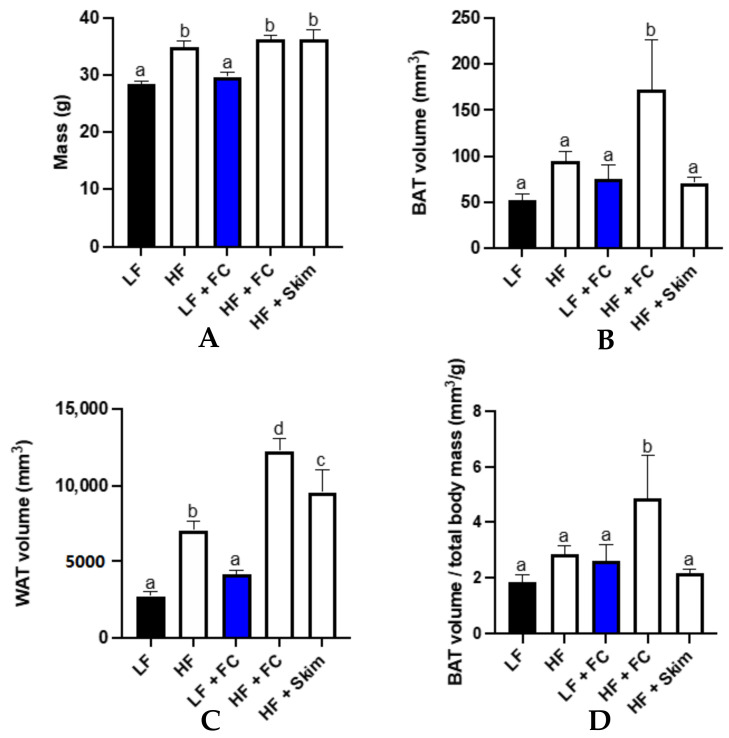
Mice body mass (**A**) and body composition analysis showing in vivo X-ray microtomography results of BAT volume (**B**) and WAT volume (**C**). BAT volume per gram of body weight (**D**) was also measured. Groups with unlike letters were significantly different (*p* < 0.05).

## Data Availability

All data are presented and available in this manuscript.

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
