# Peer review of "The Effects of Chronic Consumption of Lipid-Rich and Delipidated Bovine Dairy Milk on Brown Adipose Tissue Volume in Wild-Type Mice"

_nutrients, 2021, doi:10.3390/nu13124266_

Round 1

Reviewer 1 Report

In this manuscript, D’Alonzo et al. investigate the effect of full-cream (FC) milk on the increasing of brown adipose tissue (BAT) volume in  a mouse model fed with high-fat (HF) diet. Using X-ray microtomography, the authors reported that BAT volume and BAT volume/body mass ratio were significantly increased in HF+FC group but not mice fed with control diet or HF+skim milk. The observations are interesting and several issues need further clarification.

  1. Line 51 in Introduction, please define clear the "infant MFGM".
  2. Only male mice were studied here, is gender a factor to affect the HF+FC treatment on the BAT volume?
  3. What is the effective portion between full-cream and skim milk (lipid, protein, and/or other compounds)?
  4. In Results, please indicate figure panels with the related descriptions.
  5. For figures, some labelings of comparison pairs are confusing (for example the bottom 5 pairs in Fig. 2C). Also, please compare the sets with only one factor different. 

Reviewer 2 Report

Review of nutrients-1467188

I read a paper about “The effects of chronic consumption of lipid-rich and delipidated bovine dairy milk on brown adipose tissue volume in wild-type mice”. The HF with FC intake increases brown fat weight is interesting. However, I have a few concerns that I would like to comment on.

  1. Although the authors provide a good description of MFGM, the significance of examining the effects of commercial bovine dairy products on brown adipocyte mass is not adequately demonstrated in the introduction. Please provide How about summarizing the effects of commercial bovine dairy products on brown adipose fat mass and the relationship between MFGM and commercial bovine dairy products in the introduction?

  1. Please describe detail information of FC and skim milk used in this study. Are skim milk and FC derived from the same product? When comparing skim milk with FC, it is necessary to indicate which components in both are quantitatively and qualitatively different. However, this is not shown in detail in this paper.

  1. It's hard to understand which groups have significant differences. For example, how about annotating the groups with significant differences with different letters, and displaying a, b, c, etc. on the graph?

  1. Why can't I see the error bars in the Food intake graph?

  1. The authors suggest that browning of white adipose tissue may have occurred. but I don't think the X-ray analysis would reveal that. Are there any specific results?

  1. FC intake increases both white fat and brown fat. Are there any possible mechanisms that are common to both?

Round 2

Reviewer 1 Report

Only few points need further attention.

  1. Fig 2B and 2C are the same.
  2. Please also indicate the supplementary materials in the main text.
  3. Information in some references is missing, for example, ref 13: Nutrients . 2018 Mar 9;10(3):331.